# The Intensity of the Health Behaviors of People Who Practice Wheelchair Basketball, Wheelchair Rugby and Para-Rowing

**DOI:** 10.3390/ijerph19137879

**Published:** 2022-06-27

**Authors:** Maria Alicja Nowak, Marek Kolbowicz, Michalina Kuska, Katarzyna Sygit, Marian Sygit, Leonard Nowak, Katarzyna Kotarska

**Affiliations:** 1Institute of Physical Culture Sciences, University of Szczecin, 71-065 Szczecin, Poland; maria.nowak@usz.edu.pl (M.A.N.); marek.kolbowicz@usz.edu.pl (M.K.); leonard.nowak@usz.edu.pl (L.N.); katarzyna.kotarska@usz.edu.pl (K.K.); 2Institute of Physical Culture, Kazimierz Wielki University, 85-064 Bydgoszcz, Poland; michalinakuska@ukw.edu.pl; 3Faculty of Health Sciences, Calisia University, 62-800 Kalisz, Poland; msygit@onet.pl

**Keywords:** disabled people, Health Behavior Inventory (HBI), proper eating habits, preventive behavior, positive mental attitude, health practices

## Abstract

Sports serve people with disabilities as a form of rehabilitation. Sporting activity is a health-promoting behavior choice and a chance to achieve the best possible sports results. The aim of this study was to evaluate the intensity of health behaviors in people practicing wheelchair basketball, wheelchair rugby and para-rowing. The study encompassed 176 athletes with disabilities, aged 19 to 49 (mean age 34.41 ± 8.56), from all over Poland. Men accounted for 83.5% of the respondents. They all had a significant degree of disability and used wheelchairs in their daily lives. The authors used Juczyński’s Health Behavior Inventory (HBI) and the authors’ own survey questionnaire. Nonparametric tests were used. A greater intensity of health behaviors in general (HBI—point score) and in four categories (correct eating habits—CEH, preventive behaviors—PB, positive mental attitude—PMA, health practices—HP) was observed in rugby and basketball players (compared with para-rowers). Disabled rowers achieved the poorest results. Polish Champions scored better results in preventive behaviors (PB) and weaker results in positive mental attitude (PMA)compared with those practicing amateur sport. Respondents who trained every day and had the longest weekly exercise time (>360 min) achieved the highest results in HBI in general and in all categories compared with those who trained once a week for less than 120 min a week. Respondents with higher education, those in a better financial situation, those who were employed and married, and those who were rural residents displayed a greater intensity of health-related behaviors. People in a more difficult financial situation, who had less education, who were cohabiting, and who lived on only a pension presented more preventive behaviors. There is a need for systemic health education aimed at people with disabilities who practice various sports disciplines.

## 1. Introduction

Health behaviors are the subject of multifaceted theoretical considerations, empirical research, and various approaches to their definition and classification [1,2,3,4].

Some authors emphasize their social character, defining them as activities that directly and temporarily, or indirectly and in the longer term, favor or harm the maintenance of normative standards of physical or mental health. Selected health behaviors may be the effects of socialization or responses to social requirements or trends or constitute deliberate behaviors conditioned by knowledge and health awareness [5,6].

Prevention programs often focus on changing risky behaviors into healthy ones. Personal choices regarding the avoidance of risky behaviors (using stimulants, smoking, drinking alcohol, breaking the rules of road safety, improper diet), and implementing healthy practices (exercising, brushing teeth, environmental hygiene) as well as preventive actions (regular check-ups with a general practitioner, dentist, self-monitoring of the health condition, etc.) are of significant importance in shaping the health of society, both the able-bodied and the disabled [7,8,9,10].

Health is a condition for participation in training and achieving sports success, as well as the possibility of self-realization [10,11,12]. Sports training enables the compensation of lost functions and social integration, as well as creating opportunities for shaping pro-health choices [13,14]. A study confirmed that participation in a12-week rowing training program improved health after breast cancer [15].

The motivation of people with disabilities practicing sports does not differ in terms of striving for success and realizing their own goals and dreams in comparison with non-disabled athletes [16]. There were also no differences in motivation between able-bodied basketball players and those in wheelchairs, for whom participation in sport was also a form of rehabilitation as well as an opportunity to improve and maintain health. Increasing interest in physical activity among people with disabilities requires creating conditions and developing modern solutions in the field of rehabilitation [17,18,19].

Among the demographic and social determinants of health behaviors of various social groups, age, sex, marital status, family situation, education, profession, and financial situation are usually mentioned as factors differentiating attitudes towards health and illness, physical activity, and preferred forms of spending free time [20]. Although the implementation of pro-health behaviors is possible due to a higher economic status (easier access to medicine, sports activity, healthy diet), with insufficient health awareness and no habits, it does not ensure correct choices [21,22].

Athletes leading a healthy lifestyle are able to carry out high training loads. Playing sports requires a choice of behaviors that help achieve the desired results (e.g., avoiding smoking and drinking alcohol, eating properly).

Sports training enables the compensation of lost functions and social integration, as well as creating the possibility of shaping pro-health choices [13,14]. Training for particular sports is associated with the permanent evaluation of one’s choices, including health-promoting behaviors. Practicing sport has a great educational potential in the field of shaping social attitudes; it is a form of rehabilitation, and it also creates opportunities for social integration [23]. The physical activation of people with disabilities requires support from their family and friends in solving everyday problems [14,24,25,26,27,28]. Assistance in transporting them to trainings determines their participation in rehabilitation and sports training.

There are not many studies in the available literature that assess the intensity of health behaviors among people with disabilities practicing various sports, especially at the champion level [28].

## 2. Objective

The aim of this study was to evaluate the intensity of health behaviors among people practicing wheelchair basketball, wheelchair rugby and para-rowing.

It was hypothesized that the intensity of these athletes’ health behaviors differs depending on the sports practiced. It is also assumed that a higher level of health behavior will be presented by athletes with a high sport performance level, who exercise more often and spend more time per week\training. These dependencies may be influenced by the athlete’s level of education, material situation, employment, marital status, and place of residence.

## 3. Materials and Method

All subjects practiced paralympic sports: wheelchair basketball, wheelchair rugby, para-rowing. Athletes who take part in wheelchair basketball have various motor dysfunctions [25,29]. Rugby is a team contact combat sport that complements martial arts sports, and wheelchair rugby is “a ball game for disabled players using wheelchairs” [30], p. 585; what distinguishes rugby from individual combat sports is the team tactics. Meanwhile, para-rowing is the act of propelling a boat through water, using the muscular force of an oarsman, with their back turned in the direction of the boat’s motion. In a rowing boat, all load-bearing parts, including the axes of moving parts, must be firmly attached to the hull of the boat, except for a seat that runs on rails along the longitudinal axis [31,32]. 

Para-rowing is a group sport in which two or more athletes compete jointly with other athletes [33]. In order to qualify people for the study, the authors used the following criteria: giving written informed consent to participate in the study; having a medically significant degree of disability associated with diseases of the locomotor system; participating regularly in trainings (inclusion criteria); participating occasionally in sports activities (less than once a week); participating sports activities for less than six months; failing to specify the degree of disability; and skipping answers (exclusion criteria). Before commencing the study, the consent of the trainers, instructors, and club authorities was obtained, and the starting date was set for the study (preparatory period). The study was conducted by the authors, as well as by trainers and instructors, who were trained to do so.

The study included 176 athletes with disabilities, aged 19 to 49 (mean age 34.41 ± 8.56 years), from all over Poland. Men constituted 83.5% of the respondents. They all had a significant degree of disability and were dependent on wheelchairs in their daily lives. The respondents practiced classified (competitive) and unclassified (amateur) sports. Among the classified athletes, Polish Champions (wheelchair rugby, rowing) accounted for 29% of the respondents. The group of unclassified athletes included all wheelchair basketball athletes and some rowers (71%). Test participants most often trained 2 to 4 times a week (77.1%); 17.1% of para-rowers and 6.4% of wheelchair basketballers trained every day. The respondents spent 121 to 240 min (42.6%) on exercises per week. Wheelchair basketball players (34.6%) more often participated in sports activities for under 120 min, while rugby players (45.1%) and rowers (48.6%) more often participated in sports activities for over 240 min.

The research was carried out using the method of diagnostic survey and the standardized Health Behavior Inventory (HBI) by Juczyński as well as an original questionnaire. The HBI takes into account the frequency of individual behaviors and allows for determining the general intensity of health-promoting behaviors and the intensity of four categories of health behaviors: correct eating habits (CEH), preventive behaviors (PB), positive mental attitude (PMA), and health practices (HP). The HBI contains descriptions of 24 different health-related behaviors, and each behavior is rated on a five-point scale. The sum of the points of each category (24–120 points) indicates the intensity of the indicated behaviors. The higher the score, the greater the intensity of behavior [34].The on Cronbach’s alphas for HBI internal consistency were 0.85 for the entire HBI and 0.65, 0.61, 0.60, and 0.64, respectively, for its four subscales, CEH, PB, PMA, and HP. In our research, Cronbach’s alphas were similar for HBI (overall score) and PB (0.84 and 0.57, respectively) and higher for CEH, PMA, and HP (0.81, 0.73, and 0.68, respectively).

Individual behaviors covered four areas. Correct eating habits (CEH) mainly concerns the types of food consumed, such as vegetables, fruit, animal fats, sugars, salt, whole wheat bread. Preventive behaviors (PB) include compliance with medical recommendations, avoiding colds, knowledge of emergency numbers, and obtaining regular medical examinations. Positive mental attitude (PMA) is about avoiding too strong emotions, depressing situations, stress, and tension. Health practices (HP) include daily habits relating to rest, sleep, recreation, physical activity, maintaining proper body weight, using stimulants, and smoking.

A proprietary survey was also used to examine the style and quality of life of subjects who undertook recreational/rehabilitative or competitive physical activity. In this study, the respondents’ answers were used, including type of sports activity, weekly frequency and duration of exercises, age, sex, place of residence, marital status, education, type of professional activity, financial situation, and the degree of disability. The information obtained from the respondents was supplemented and verified by interviews with their trainers, instructors, and rehabilitators (how long the respondents practiced sports, how regularly they trained, their degree of disability, consent for survey, and appointing the time frames for the study).

Written informed consent was obtained from each subject included in the study. The research was approved by the Bioethics Committee at the Regional Medical Chamber in Szczecin No. 15/KB/V/2015. We conducted the research in accordance with the Declaration of Helsinki of 1975.

### Statistical Analysis

After examining the normality of the distribution (deviating from normal), nonparametric statistics were applied in the analyses of the results. The Kruskal–Wallis test (H) was used to compare several independent samples. In the case of determining the statistical significance of the differences in the comparisons of two independent samples, the Mann–Whitney (U) test was employed. The independence chi-square test was used to examine the relationships between two variables. The effect size was calculated for each test: E^2^_R_ for the Kruskal–Wallis H test, Glass rank biserial correlation (rg) for the Mann–Whitney U test, and Cramér’s V for the χ^2^ test [29]. The value of *p* < 0.05 was assumed to be statistically significant. Statistical calculations were made with Statistica 13.1 for Windows (StatSoft Sp. zo.o., Krakow, Poland) and Microsoft Office Excel 2007 (Microsoft Sp. z o.o., Warsaw, Poland).

## 4. Results

### 4.1. The Socio-Economic Characteristics

The respondents were 19 to 49 years old, with most being under 40 (75%). Women were a minority (16.5%). City dwellers predominated (85.2%) with the exception of rugby players, among whom there were more members int he rural population (51.6%) (*p* < 0.001 for χ^2^; Cramér’s V = 0.6). Over half of respondents, 52.3%, were in formal relationships. Wheelchair rugby and para-rowing athletes were more often unmarried than were wheelchair basketball players (*p* = 0.005 for χ^2^). Almost half of the respondents had higher education, the majority of whom were basketball players (61.8%) (*p* < 0.001 for χ^2^; Cramér’s V = 0.3). Most of the respondents worked professionally, most often wheelchair basketball players (78.2%). Students who combined work and university accounted for 15.5%. Rowers more often lived off pension (34.3%) (*p* = 0.002 for χ^2^). The respondents assessed their financial situation mainly as good (51.7%). Wheelchair basketball players rated their financial situation the highest (39.1%), while rugby players and para-rowers rated theirs lower (35.5% and 37.1%, respectively) (*p* < 0.001 for χ^2^; Cramér’s V = 0.3) (Table 1).

All wheelchair rugby players and 42.9% of the para-rowers won the title of the Polish Champions. The wheelchair basketball players practiced amateur sports. The respondents most often trained 2 to 4 times a week (77.1%), with 17.1% of para-rowers and 6.4% of wheelchair basketballers training every day. The respondents spent 121 to 240 min (42.6%) on exercise per week. The wheelchair basketball players (34.6%) more often participated in sports activities under 120 min, while rugby players (45.1%) and para-rowers (48.6%) more often participated in sports activities over 240 min (*p* < 0.001 for χ^2^; Cramér’s V = 0.3).

### 4.2. The Health Behavior Intensity of Basketball and Rugby Players in Wheelchairs as Well as Para-Rowers Depending on the Sport Performance Level, Frequency and Time of Exercise

There was a general differentiation in the intensity of health behaviors among people practicing wheelchair basketball, wheelchair rugby and para-rowing (*p* < 0.05) (H test) (Table 2). A better overall score in HBI (point score) was achieved by rugby players and wheelchair basketball players compared with para-rowers (*p* < 0.001 and *p* = 0.007 for the U test, respectively) with high and average effect size (rg = 0.6 and rg = 0.3, respectively). Correct eating habits (CEH) were more common among basketball players and rugby players than para-rowers (*p* = 0.001; *p* = 0.006).

As far as prophylactic behavior was concerned, there were better results for the basketball players and weaker ones for the rugby players and para-rowers (*p* < 0.001 and for the U test; rg = −0.7 and rg = −0.4, respectively). The rugby players presented a higher level of preventive behavior in comparison with the para-rowers (*p* = 0.003 for the U test; rg = 0.4). Better results in positive mental attitude (PMA) were achieved by the basketball players compared with the rugby players and para-rowers (*p* = 0.016 and *p* < 0.001 for the U test, respectively) as well as the rugby players compared with the para-rowers (*p* = 0.004 for the U test). The basketball and wheelchair rugby players chose health practices (HP) more often than the para-rowers (*p* = 0.008 and *p* < 0.001 for the U test, respectively). The relationships within PMA and HP showed average and high effect sizes (rg = 0.3–0.5).

Amateurs (A) less frequently presented preventive behaviors (PB) and more often presented positive mental attitude (PMA) compared with the Polish Champions (PC) (*p* < 0.001; *p* = 0.005 for the U test). The relationships within PB and PMA showed average and great effect size (rg = 0.3 and 0.6 respectively) (Figure 1).

The best results in HBI and in the CEH, PMA and HP categories were reported among people with the longest weekly exercise time (>360 min), while the worst ones were among people exercising less than 120 min (*p* = 0.018, *p* = 0.014, *p* = 0.013, and *p* = 0.009 for the U test, respectively). People who devoted less time to exercise also achieved poorer results (PMA, HP) than those who exercised 121–240 min (*p* = 0.006 and *p* = 0.037 for the U test, respectively). These differences are confirmed by the mean and effect size (rg = −0.3; 0.4) above the mean. People exercising less than 120 min a week had better results in PB than did those who exercised 121–240min (*p* = 0.035 for the U test), who in turn had weaker results when compared with those who exercised 241–360min (*p* = 0.035 for the U test) (Table 3).

A general differentiation in the intensity of health behaviors (PB, CEH) of the respondents who exercised at different weekly frequencies (*p* < 0.05) (H test) was observed (Table 4). Pairwise comparisons show differences in HBI (point score), CEH, and PMA between those exercising daily and exercising once a week (*p* = 0.035; *p* = 0.032; *p* = 0.032 for the U test), as well as exercising twice a week and 3–4 times in the PB range (*p* = 0.014; *p* = 0.011 for the U test). These differences are confirmed by the effect size above the mean (rg = 0.4). The respondents who trained daily achieved the highest results in HBI and in all categories. Basketball, rugby and para-rowing athletes who practiced weekly had poorer results in CEH compared with those exercising twice a week (*p* = 0.036 for the U test) as well as those exercising 3–4 times a week in PMA (*p* = 0.011 and *p* = 0.007 for the U test, respectively) and HP (*p* = 0.028 and *p* = 0.035 for the U test, respectively). These differences are confirmed by the effect size (rg = −0.3, rg = −0.4). Those exercising once a week achieved the lowest scores in HBI and all categories.

### 4.3. The Intensity of Health Behaviors of People with Disabilities Practicing Sports Depending on Their Education, Material Situation, Work, Marital Status and Place of Residence

The general differentiation between education level and HBI and its categories was confirmed by the H test (*p* < 0.05) (Table 5). Comparison of the results in pairs indicates better results for respondents with secondary and higher education compared with those with lower secondary education in the field of HBI (point score) (*p* = 0.004 and *p* = 0.031 for the U test, respectively), CEH (*p* < 0.001 and *p* = 0.002 for the U test, respectively), PMA (*p* < 0.001 and *p* < 0.001 for the U test, respectively) and HP (*p* < 0.001 and *p* = 0.018 for the U test, respectively) (effect size ranging from rg = −0.2 to −0.6). Only in PB were better results achieved by people with secondary and below secondary education (*p* = 0.003 and *p* < 0.001 for the U test, respectively; rg = 0.04).

Differences in preventive behaviors (PB) between working people, students, and pensioners were observed (*p* = 0.004; *p* = 0.040 for the U test, respectively). Effect size: rg = −0.3; rg = −0.4 (Table 6).

In this category, the best results were achieved by pensioners. There was a general differentiation in the intensity of the health behaviors of the respondents depending on their self-assessment of their financial situations (test H) (*p* < 0.05) (Table 7). Comparisons in pairs indicate better results for respondents in a very good financial situation compared with those who assessed their situation as good in HBI, CEH, PMA, and HP (*p* = 0.009, *p* = 0.009, *p* < 0.001, and *p* = 0.001 for the U test, respectively). Effect sizes ranged from 0.3 to 0.4. Better results were obtained in all HBI categories (CEH, PB, PMA, and HP) by people who assessed their situation good compared with those who assessed it as sufficient (*p* = 0.001; *p* < 0.001; *p* < 0.001; *p* = 0.005 for the U test, respectively). High effect sizes were found in PB and PMA between the very good and sufficient self-assessments of the respondents’ financial situations (rg = −0.6; rg = 0.5).

General differentiation was confirmed in HBI, CEH, PMA, and HP among people with different marital status (H test) (*p* < 0.05) (Table 8). Married people outperformed the unmarried in HBI, CEH, and PMA (*p* = 0.026, *p* = 0.017, and *p* = 0.013 for the U test, respectively) and compared with divorced people (*p* = 0.013, *p* = 0.019, and *p* = 0.003 for the U test, respectively). The poorest results in HBI, CEH, PMA, and HP were achieved by people living in cohabitation compared with those who were married (*p* = 0.008, *p* = 0.029, *p* = 0.001, and *p* = 0.004 for the U test, respectively) and in PMA and HP (*p* = 0.017 and *p* = 0.025 for the U test, respectively) for those who were unmarried. High effect sizes were found in comparisons between married and single with cohabiting people in HBI and its categories (rg = 0.4 to 0.7).

Athletes living in the countryside had better results in PB compared with those living in the city (*p* = 0.001 for the U test, rg = −0.4), who achieved better results in PMA (*p* = 0.022 for the U test, rg = 0.3) (Table 9).

## 5. Discussion

The aim of this study was to evaluate the intensity of health behaviors in people practicing wheelchair basketball, wheelchair rugby and para-rowing. The study confirmed the authors’ hypothesis that the intensity of health behaviors of respondents differed depending on their sports discipline. A better overall score in HBI was achieved by rugby players and wheelchair basketball players compared with para-rowers. Correct eating habits (CEH) and health practices (HP) were more common among both basketball and rugby players (compared with para-rowers). Better results in positive mental attitude (PMA) and preventive behaviors (PB) were achieved by basketball players compared with rugby players and para-rowers and by rugby players compared with para-rowers. The para-rowers scored the lowest in each HBI category. These results may be explained by the better education of wheelchair basketball players (61.8% had higher than secondary education), which gave them awareness in various areas of life (including health education). The obtained results are aligned with other studies [35,36,37]. Wheelchair basketball players had better results in dietary habits, preventive behaviors, health practices, and positive mental attitude compared with other sports practitioners. Positive mental attitude plays a significant motivating role and influences the experiences and actions of athletes. High mean indicators were also obtained for eating habits, which is also important for athletes, because the rational way of their nutrition influences the improvement of exercise capacity and improvement of the players’ health potential [36,37]. It is emphasized in the literature that people who practice sports usually pay attention to proper nutrition, avoid stimulants, rest regularly, and choose forms of leisure adequate to their needs [37].

In their own research, the authors discovered better results among people with disabilities who practiced team sports (wheelchair basketball and rugby) in terms of correct eating habits (CEH) and health practices (HP). The authors believe that they resulted from cooperation of respondents with a dietitian (sometimes a full-time club employee). Para-rowers cannot use the help of a dietitian, mainly for financial reasons. This problem could be partially solved by raising para-rowers’ awareness of nutrition and health practices, by organizing trainings, lectures, meetings with a dietitian, etc. [38,39,40]. In the light of the literature, the correlation between the awareness and diet of athletes is ambiguous. The low quality of research and the heterogeneity of methods make it impossible to draw precise conclusions. Proven tools for measuring awareness and its impact on diets need to be developed. By monitoring and modifying their diet, respondents may avoid obesity related to spinal cord injury, as well as maintain a correct body weight which would help them achieve positive sports results [16,41].

The assumption about higher intensity of health behavior among subjects with a high level of sports performance was partially confirmed. However, no differences were observed in terms of HBI, CEH, and HP between amateurs and Polish Champions. Polish Champions had better results in preventive behaviors (PB), while amateurs scored better in terms of positive mental attitude (PMA). Differences in behaviors may result from the requirement for systematic medical check-ups faced by athletes who take part in competitions. It would be necessary for the trainers to have a greater influence on shaping healthy behaviors of their disabled trainees [42,43,44]. In this study, the authors found that a higher intensity of health behaviors was displayed by subjects who exercised more frequently and spent more time per week on training. The respondents who trained every day achieved the highest results in the examined health behaviors (HBI and its categories). Compared with those exercising once a week, they had better results in terms of correct eating habits (CEH) and positive mental attitude (PMA).The best results in HBI and its categories (CEH, PMA, HP) were obtained by people with the longest weekly exercise time (>360 min), while the weakest ones were obtained by those exercising for less than 120 min. These results are partially confirmed by studies of able-bodied athletes and martial artists. Correct eating habits and health practices were typical of those who exercised daily [45] The number of training sessions (5 times a week) of judo practitioners was one of the factors determining the increase in their intensity of healthy behaviors [46].

The authors confirmed that the intensity of health behaviors of people who practiced wheelchair basketball, wheelchair rugby and para-rowing was connected with their education, financial situation, employment, marital status, and place of residence. Respondents with secondary and higher education scored better results in health behaviors (HBI point score, CEH, PMA and HP) compared with those with elementary education; the same trend was observed between respondents in very good and good financial situations compared with those assessing their situations as adequate. Married people outperformed those who were single (in terms of HBI, CEH, and PMA). Marriage and having a family had a positive impact on the practice of sports by people with disabilities. By accepting a subject’s sports activity, the family created conditions for practicing their chosen discipline [47]. The authors observed better results in the preventive behaviors (PB) of the respondents with secondary and elementary education, those in a worse financial situation, and among pensioners. The available studies confirmed that disabled athletes coped better with the difficulties of everyday life, participated in various areas of life, and were more likely to be employed [48] A higher level of preventive behaviors (PB) was also displayed by disabled athletes who lived in the countryside. More positive preventive behaviors were also seen among students from rural areas [49,50]. Significant changes may be noticed in the level of society’s interest in sports of people with disabilities. Paralympic Games are fully broadcast in the media. Many municipalities allocate financial merit-based rewards for participants of the Paralympic Games at the same level as for able-bodied athletes. It is an important step in familiarizing the society with the situation of people with disabilities, their sports achievements, showing the joy of competitiveness and victory in martial arts, the ability to overcome their health problems, self-acceptance, and their desire to be accepted and respected by the society.

## 6. Conclusions

There is a need for systemic health education aimed at people with disabilities who practice various sports disciplines.

## 7. Limitations

All subjects practiced paralympic sports. The adopted framework resulted in a limited number of respondents, especially women. The authors plan to fill this gap in future research projects. The poorest results in health behaviors were obtained by para-rowers. This group was dominated by solo rowers. Including a larger number of respondents (e.g., team rowers, athletes practicing individual sports, women, trainers, physiotherapists and sports nutritionists) would allow for a more precise assessment of preferred health behaviors and the development of adequate methods of working with athletes with disabilities.

## Figures and Tables

**Figure 1 ijerph-19-07879-f001:**
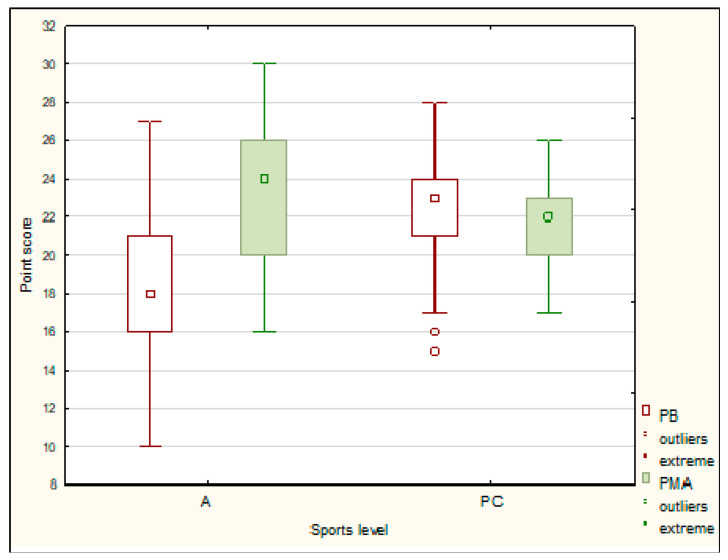
The health behaviors of the Polish Champions (PC) and the amateurs (A).

**Table 1 ijerph-19-07879-t001:** Socioeconomic characteristics of people practicing wheelchair basketball, wheelchair rugby and para-rowing (independence χ^2^ test and Cramér’s V).

Variables	Sport			Total (176)		*p* for χ^2^	Cramér’s V
	Wheelchair Basketball (*n* = 110)	Wheelchair Rugby (*n* = 31)	Para-Rowing (*n* = 35)	N	%		
Sex: -man -woman	80.9 19.1	100.0 -	77.1 22.9	147 29	83.5 16.5	-	-
Age: ≤29 years 30–39 ≥40	36.4 40.9 22.7	19.4 54.8 25.8	40.0 28.6 31.4	60 72 44	34.1 40.9 25.0	n.s.	-
Place of residence: -city -village	99.1 0.9	48.4 51.6	74.3 25.7	150 26	85.2 14.8	0.000	0.6
Relationship status: -I am not in a relationship -married -divorced -I have a partner	21.8 62.7 10.9 4.5	48.4 38.7 9.7 3.2	57.1 31.4 2.9 8.6	59 92 16 9	33.5 52.3 9.1 5.1	0.005	0.2
Education: -below secondary -secondary -above secondary	14.6 23.6 61.8	12.9 58.1 29.0	22.9 51.4 25.7	28 62 86	15.9 35.2 48.9	0.000	0.3
Job type: -pupil/student -work -studying and working -disability pension	7.3 78.2 4.5 10.0	9.7 58.0 9.7 22.6	20.0 40.0 5.7 34.3	18 118 10 30	10.2 67.0 5.7 17.1	0.002	0.2
Financial situation: -very good -good -sufficient	39.1 52.7 8.2	3.2 61.3 35.5	22.9 40.0 37.1	52 91 33	29.6 51.7 18.7	0.000	0.3

**Table 2 ijerph-19-07879-t002:** Health behaviors of people practicing wheelchair basketball, wheelchair rugby and rowing (the H test, E^2^_R_, the U test, rg).

HBI and Its Categories	Sports	Wheelchair Rugby	Para-rowing	Wheelchair Rugby	Para-Rowing	Rank Means
		Value of *p* for U Statistics		Glass Rank Biserial Correlation (rg)		
**HBI** (point score) H(2,176) = 11.65 E^2^_R_ = 0.1 *p* = 0.003 *	Wheelchair basketball	0.516	0.007 *	−0.1	0.3	92.56
	Wheelchair rugby		0.000 *		0.6	102.54
	Para-rowing					63.27
**CEH** H(2,176) = 12.08 E^2^_R_ = 0.1 *p* = 0.002 *	Wheelchair basketball	0.332	0.001 *	0.1	0.4	96.65
	Wheelchair rugby		0.006 *		0.4	89.03
	Para-rowing					62.41
**PB** H(2,176) = 46.94 E^2^_R_ = 0.3 *p* = 0.000 *	Wheelchair basketball	0.000 *	0.000 *	−0.7	−0.4	96.65
	Wheelchair rugby		0.003 *		0.4	89.03
	Para-rowing					62.41
**PMA** H(2,176) = 23.42 E^2^_R_ = 0.1 *p* = 0.000 *	Wheelchair basketball	0.016 *	0.000 *	0.3	0.5	101.53
	Wheelchair rugby		0.004 *		0.4	80.12
	Para-rowing					54.95
**HP** H(2,176) = 10.98 E^2^_R_ = 0.1 *p* = 0.004 *	Wheelchair basketball	0.624	0.008 *	−0.1	0.3	92.80
	Wheelchair rugby		0.000 *		0.5	101.09
	Para-rowing					63.82

* statistically significant for *p* < 0.05.

**Table 3 ijerph-19-07879-t003:** Health behaviors of people practicing wheelchair basketball, wheelchair rugby and para- rowing depending on the weekly exercise time (the H test, E^2^_R_, the U test, rg).

HBI and Its Categories	Weekly Exercise Time	121–240	241–360	>360	121–240	241–360	>360	Rank Means
		Value of *p* for U Statistics			Glass Rank Biserial Correlation (rg)			
**HBI** (point score) H(3,176) = 6.31 E^2^_R_ = 0.04 *p* = 0.097	<120	0.194	0.404	0.018 *	−0.1	−0.1	−0.4	77.93
	121–240		0.799	0.078		0.1	−0.3	90.06
	241–360			0.058			−0.3	87.20
	>360							114.71
**CEH** H(3,176) = 7.49 E^2^_R_ = 0.04 *p* = 0.058	<120	0.093	0.308	0.014 *	−0.2	−0.1	−0.4	75.85
	121–240		0.495	0.119		0.1	−0.2	92.21
	241–360			0.040 *			−0.4	85.70
	>360							114.25
**PB** H(3,176) = 6.65 E^2^_R_ = 0.04 *p* = 0.084	<120	0.035 *	0.723	0.414	0.2	−0.1	0.1	97.29
	121–240		0.035 *	0.593		−0.2	−0.1	77.86
	241–360			0.377			0.2	99.79
	>360							85.53
**PMA** H(3,176) = 10.56 E^2^_R_ = 0.06 *p* = 0.014 *	<120	0.006 *	0.065	0.013 *	−0.3	−0.2	−0.4	69.52
	121–240		0.364	0.395		0.1	−0.1	96.24
	241–360			0.151			−0.2	87.98
	>360							108.03
**HP** H(3,176) = 9.05 E^2^_R_ = 0.05 *p* = 0.028 *	<120	0.037 *	0.093	0.007 *	−0.2	−0.2	−0.4	72.34
	121–240		0.808	0.110		0.0	−0.3	92.20
	241–360			0.104			−0.3	90.07
	>360							113.71

* statistically significant for *p* < 0.05.

**Table 4 ijerph-19-07879-t004:** Health behaviors of people practicing wheelchair basketball, wheelchair rugby and para-rowing depending on the frequency of exercise per week (the H test, E^2^_R_, the U test, rg).

HBI and Its Categories	The Frequency of Exercises per Week	Once a Week	2 Times a Week	3–4 Times a Week	Once a Week	2 Times a Week	3–4 Times a Week	Rank Means
		Value of *p* for U Statistics			Glass Rank Biserial Correlation (rg)			
**HBI** (point score) H(3,175) = 5.09 E^2^_R_ = 0.03 *p* = 0.165	every day	0.035 *	0.240	0.171	0.4	0.2	0.2	108.80
	once a week		0.101	0.199		−0.2	−0.2	72.18
	2 times a week			0.656			0.1	91.04
	3–4 times a week							87.18
**CEH** H(3,175) = 6.09 E^2^_R_ = 0.04 *p* = 0.107	every day	0.032 *	0.407	0.336	0.4	0.1	0.2	104.34
	once a week		0.036 *	0.068		−0.3	−0.2	67.79
	2 times a week			0.782			0.0	91.68
	3–4 times a week							89.19
**PB** H(3,176) = 9.91 E^2^_R_ = 0.06 *p* = 0.019 *	every day	0.263	0.014 *	0.011 *	0.2	0.4	0.4	120.69
	once a week		0.072	0.060		0.2	0.2	102.92
	2 times a week			0.699			0.0	83.28
	3–4 times a week							80.38
**PMA** H(3,175) = 8.98 E^2^_R_ = 0.05 *p* = 0.029 *	every day	0.032 *	0.637	0.936	0.4	0.1	0.0	97.11
	once a week		0.011 *	0.007 *		−0.3	−0.4	61.92
	2 times a week			0.517			−0.1	89.83
	3–4 times a week							94.95
**HP** H(3,175) = 5.67 E^2^_R_ = 0.03 *p* = 0.128	every day	0.117	0.893	0.795	0.3	0.0	0.0	94.57
	once a week		0.028 *	0.035 *		−0.3	−0.3	66.81
	2 times a week			0.945			0.0	91.97

* statistically significant for *p* < 0.05.

**Table 5 ijerph-19-07879-t005:** Health behaviors of people practicing wheelchair basketball, wheelchair rugby and rowing depending on the level of education (the H test, E^2^_R_, the U test, rg).

HBI and Its Cat	Level of Education	Secondary	Above Secondary	Secondary	Above Secondary	Rank Means
		Value of *p* for U Statistics		Glass Rank Biserial Correlation (rg)		
**HBI** (point score) H(2,176) = 10.10 E^2^_R_ = 0.06 *p* = 0.006 *	Below secondary	0.214	0.004 *	−0.2	−0.4	67.76
	Secondary		0.031 *		−0.2	81.86
	Above secondary					100.03
**CEH** H(2,176) = 17.84 E^2^_R_ = 0.10 *p* = 0.000 *	Below secondary	0.235	0.000 *	−0.2	−0.5	63.44
	Secondary		0.002 *		−0.3	77.91
	Above secondary					104.29
**PB** H(2,176) = 17.79 E^2^_R_ = 0.10 *p* = 0.000 *	Below secondary	0.520	0.003 *	−0.1	0.4	101.94

* statistically significant for *p* < 0.05.

**Table 6 ijerph-19-07879-t006:** Health behaviors of people practicing wheelchair basketball, wheelchair rugby and para- rowing depending on the work performed (the U test, rg).

HBI and Its Categories	Work Type	Work	Studying and Working	Disability Pension	Work	Studying and Working	Disability Pension	Rank Means
		Value of *p* for U Statistics			Glass Rank Biserial Correlation (rg)			
**PB** H(3,176) = 9.96 E^2^_R_ = 0.06 *p* = 0.019 *	pupil/student	0.382	0.248	0.299	0.1	0.3	−0.2	94.69
	work		0.293	0.004 *		0.2	−0.3	83.21
	studying and working			0.040 *			−0.4	67.60
	disability pension							112.53

* statistically significant for *p* < 0.05.

**Table 7 ijerph-19-07879-t007:** Health behaviors of people practicing wheelchair basketball, wheelchair rugby and para-rowing depending on their financial situation (the H test, E^2^_R_, the U test, rg).

HBI and Its Categories	Financial Situation	Good	Sufficient	Good	Sufficient	Rank Means
		Value of *p* for U Statistics		Glass Rank Biserial Correlation (rg)		
**HBI** (point score) H(2,176) = 7.23 E^2^_R_ = 0.04 *p* = 0.027 *	Very good	0.009 *	0.065	0.3	0.2	104.38
	Good		0.814		−0.0	81.21
	Sufficient					83.56
**CEH** H(2,176) = 9.31 E^2^_R_ = 0.05 *p* = 0.009 *	Very good	0.009 *	0.001 *	0.3	0.3	105.99
	Good		0.428		0.1	83.18
	Sufficient					75.60
**PB** H(2,176) = 21.22 E^2^_R_ = 0.12 *p* = 0.000 *	Very good	0.003 *	0.000 *	−0.30	−0.6	64.95
	Good		0.026 *		−0.3	92.02
	Sufficient					115.89
**PMA** H(2,176) = 21.31 E^2^_R_ = 0.12 *p* = 0.000 *	Very good	0.000 *	0.000 *	0.4	0.5	115.53
	Good		0.621		0.1	78.75
	Sufficient					72.77
**HP** H(2,176) = 12.69 E^2^_R_ = 0.07 *p* = 0.002 *	Very good	0.001 *	0.005 *	0.3	0.4	109.50
	Good		0.847		0.0	80.26
	Sufficient			0.3	0.2	78.10

* statistically significant for *p* < 0.05.

**Table 8 ijerph-19-07879-t008:** Health behaviors of people practicing wheelchair basketball, wheelchair rugby and para- rowing depending on marital status (the H test, E^2^_R_, the U test, rg).

HBI and Its Categories	Marital Status	Married	Divorced	Cohabitating	Married	Divorced	Cohabitating	Rank Means
		Value of *p* for U Statistics			Glass Rank Biserial Correlation (rg)			
**HBI** (point score) H(3,176) = 14.09 E^2^_R_ = 0.081 *p* = 0.003 *	Single	0.026 *	0.236	0.071	−0.2	0.2	0.4	81.83
	Married		0.013 *	0.008 *		0.4	0.5	100.41
	Divorced			0.630			0.1	65.34
	Cohabitating							51.55
**CEH** H(3,176) = 11.88 E^2^_R_ = 0.068 *p* = 0.008 *	Single	0.017 *	0.373	0.267	−0.2	0.1	0.2	80.15
	Married		0.019 *	0.029 *		0.4	0.4	100.20
	Divorced			0.820			0.1	67.53
	Cohabitating							60.88
**PMA** H(3,176) = 20.18 E^2^_R_ = 0.115 *p* = 0.000 *	Single	0.013 *	0.077 *	0.017 *	−0.2	0.3	0.5	82.14
	Married		0.003 *	0.001 *		0.5	0.7	102.09
	Divorced			0.276			0.3	60.16
	Cohabitating							41.72
**HP** H(3,176) = 12.09 E^2^_R_ = 0.069 *p* = 0.007 *	Single	0.068	0.323	0.025 *	−0.2	0.2	0.5	83.80
	Married		0.068	0.004 *		0.3	0.6	98.56
	Divorced			0.332			0.2	71.62
	Cohabitating							46.38

* statistically significant for *p* < 0.05.

**Table 9 ijerph-19-07879-t009:** Health behaviors of people practicing wheelchair basketball, wheelchair rugby and para-rowing depending on place of residence (the U test, rg).

HBI and Its Categories	Place of Residence	Countryside	Countryside	Rank Means
		Value of *p* for U Statistics	Glass Rank Biserial Correlation (rg)	
**PB**	City	0.001 *	−0.4	83.15
	Countryside			119.37
**PMA**	City	0.022 *	0.3	92.16
	Countryside			67.38

* statistically significant for *p* < 0.05.

## Data Availability

Not applicable.

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
