# Peer review of "The Intensity of the Health Behaviors of People Who Practice Wheelchair Basketball, Wheelchair Rugby and Para-Rowing"

_ijerph, 2022, doi:10.3390/ijerph19137879_

Round 1

Reviewer 1 Report

The aim of the study was to evaluate the intensity of health behavior of people practicing different wheelchair sports.  The article is interesting and presents good results. However, minor changes are needed to improve readability.

·      it is not clear how the participants were recruited.

·      The three sports considered have many differences. How is this considered?

·      Have exclusion criteria been taken into account?

·      Diseases affect physical performance. How has this been taken into account?

·      Figure 2 is unclear.

Reviewer 2 Report

The research is of interest and addresses a demographic in need, however, the manuscript is poorly presented and the summary findings are not in-line with what was tested. It is also unclear if all the conclusions are appropriate for cross-sectional data or if more conservative statistical methods would be more appropriate. Multiple-comparisons and potential interactions are also not tested statistically. While it is possible that it can be reworked for successful publication, at present it is unacceptable. Specific comments for improvement are provided below.

Throughout the manuscript – avoid the abbreviations HBI, CEH, PB, PMA and HP. If you opt to use them in the tables and figures, please define them. The terms are not standardized and given the large number of outcomes and comparisons it is difficult to constantly refer back to these to interpret the text. Please use the full-term unless it is absolutely not possible.

Throughout the manuscript – many sentences have a “.” Not followed by a space, please check.

Abstract

Line 19 – expand; which non-parametric stats, provide more detail? Statement regarding p-value in abstract can be removed as this is assumed.

Line 27 and throughout – You cannot assume a direction with this type of data in many cases where you do. Chi squared tests only allow you to speak about relationships. This is a significant flaw throughout the manuscript and needs to be corrected. Furthermore, often correlational statistics would be more appropriate as there was no intervention. For example, how can you be sure that health behaviours are determined by training volume versus health behaviours impacting people’s training. It is possible that people with higher levels of healthier behaviours are physically healthier and thus able to train more. It is more appropriate to say that there is a relationship between the variables.

Line 30 – while this is a likely true and valuable comment regarding sports activity; your research does not address this as you were not making comparisons between athlete and non-athlete groups.

Introduction

Line 48 – please explain what the “rules of communication safety are”. I don’t believe the term is widely understood.

Line 62 – please explain why the needs of people with disabilities required greater public attention?

Line 71 – Sentence starting with “Training…” requires rewording as it isn’t clear.

Line 75 – Statement regarding socio-cultural activation requires additional information regarding the “supports” if it is maintained. However, given that the manuscript focuses on personal determinants of health and not policy I’m not sure it is relevant. I would suggest focusing the introduction more on the personal determinants of health you are measuring and research in athletes with disabilities.

Line 78 - This is confusing; health is discussed more broadly throughout the introduction and then suddenly it is about falls or collisions with vertical obstacles. Furthermore, why are those with increased risk of falls or collisions mentioned? They are not included in the research.

Objective

Line 84 – for clarity please specify that it is athletes in wheelchairs so it is clear that the “rowers” are not non-disabled rowers.

While the hypothesis regarding the different sports is interesting, there is no indication of why those sports were selected. What is unique or different about them that could explain the difference in results. Usually we classify sports as endurance, strength, intermittent, team, individual etc.

Methods

Line 100 – what is the original questionnaire is it the HBI referred to earlier? This is confusing? If possible, please attach a copy of the questionnaire(s) in a supplemental file or link.

Information on how individuals were recruited, what the inclusion and exclusion criteria were, in what format and scenario the athletes completed the questionnaire, training phase etc. is required.

Line 118 – what was the original survey technique?

Line 119 – what were these interviews? What was asked? How were they conducted? Who conducted them?

Was a correction used for multiple comparisons?

Line 133 – none of your statistics are qualitative; there is no such thing.

Results

Throughout – a p-value that is truly “0” is not possible; p=0.000 should be written as p<.001.

Throughout - As previously mentioned you cannot have a directional hypothesis with a chi-square test. A chi-square can only tell you that a relationship exists between the two variables not which factor is influencing the other. I would also confirm that the other statistical tests are appropriate for cross-sectional data. I believe looking at correlations would be more appropriate.

Table 1 – there does not appear to be a financial situation option for less than sufficient; however, it is often the case that incomes of those with a spinal cord injury are below their needs? Why was this not an option?

Figure 1 – Doesn’t make sense; there is a percent sign in the y-axis although it should indicate a percent of what; also it goes above 100. There are also numbers in the bars that don’t correspond with the axis?

Tables are often not clear which comparisons the p-values refer to. For the pairwise comparisons please use different symbols to indication comparisons between differing groups of two.

Figure 2 - Chart is squished and difficult to read. y-axis requires a label.

Discussion

Overall – I have concerns with the direction of the conclusions. The data is cross-sectional. I think it is equally likely that the health behavior intensity is influencing the duration that people can exercise, work, study etc.

Line 277-285 – What are some possible explanations about the difference between rowers and other sports? What is different about the sports?

Line 291 – This statement is confusing as it implies it is harder for athletes to make healthy choices and contradicts the other statements.

 Line 295 – In this part of the discussion, it should stay focused on the difference between the athlete groups as you did not make comparisons to non-athletes.

Line 305 – It would be valuable to cite studies that report sport nutrition knowledge levels in athletes with disabilities.

Line 307 - Why do team sports have a dietitian? Also, why was access to a dietitian not studied in this research. It seems relevant to questions regarding healthy eating behaviours. Has this other research been published?

Line 311 – please be cautious with this statement. While it is mentioned that other people were interviewed that data is not presented in the results. Did you actually question the rowers regarding their access to dietitians? Or only the coaches and authorities? Regardless the data needs to be presented if it is going to be discussed in the results.

Line 316-320 – References required to support these statements.

Line 321 - Why? Is there other research to support this theory?

Line 325-331 – References required to support this statement.

Line 335 – Again, I’m not sure the relationship isn’t reversed? What evidence do you have that health behavior intensification isn’t affecting exercise levels?

Line 356 - Can you confirm that the athletes were training with the trainer and not on their own? What percentage of the time did they train with a trainer vs a group. Did you ask about contact time with athlete support personnel? Also, what phase of training were the athletes in. This is relevant to training volume statements and likely their HBI.

Please consider confounding factors in your analysis. Multiple regression techniques might be more appropriate. For example, perhaps those who are married also have increased income as compared to those who are single. I’m not sure all of these factors should be analyzed individually?

Line 403 – Why is the statement regarding family being more important to women relevant to your study?

Line 410-417 – This paragraph is not relevant to your study as you did not study the benefits of physical activity vs sedentary in your data sets. While the statement is likely true; nothing in your research studies the benefit to society of para-sport. The best you can do is talk about the different sports.

Conclusion – needs to be re-written to be relevant to the current study (see comment above).

Line 426 – confusing as you mentioned rowers are solo and now you are discussing team and group sports. If you want to center your manuscript around the team vs. groups sports please define this and explains which sports are which and why this was a study outcome. Address it in your introduction and throughout.

Limitations – presumably this is self-report questionnaire data, which has significant limitations. I would be valuable in the methods to outline how the data was collected? How were they recruited? How did they complete the questionnaire? What controls were in place; you can then address any gaps in the limitations.  

Reviewer 3 Report

The present study examines the link between health-related behaviors and the intensity of sports participation among people with disabilities in Poland. The sample included 176 athletes with disabilities aged 19 to 49 (males constituted 83.5% of the sample) practicing wheelchair basketball, wheelchair rugby, and rowing. Health-related behaviors were assessed using Health Behavior Inventory (HBI), Correct Eating Habits (CEH), Preventive Behaviour (PB), Positive Mental Attitude (PMA), and Health Practices (HP). Results show that all these aspects of health behaviors (HBI, CEH, PB, PMA) were influenced by a higher level of education, better financial situation, taking up employment, marital status and place of residence. The authors conclude that active engagement in sports has many benefits for the rehabilitation of people with physical disabilities, and athletes leading a healthy lifestyle are able to rehabilitate faster.

The study is scientifically well-grounded, and the analyses make sense. However, the analyses are based on a small, selective, and unrepresentative sample. Moreover, the study sample is dominated by men (males outnumber females greater than four to one). All this impedes the generalization of the findings. Moreover, the study is based on a cross-sectional design. This shortcoming limits the ability to draw any causal implications.

Round 2

Reviewer 3 Report

The authors have managed to deal with the reviewer's concerns in the proper way, and I believe that the current version of the manuscript deserves to be published.